# *Toxoplasma* infection in male mice alters dopamine-sensitive behaviors and host gene expression patterns associated with neuropsychiatric disease

**Graham L. Cromar**[1☉], **Jonathan R. Epp**[2☉¤], **Ana Popovic**[1,3], **Yusing Gu**[2], **Violet Ha**[2], **Brandon J. Walters**[2], **James St. Pierre**[1], **Xuejian Xiong**[1], **John G. Howland**[4], **Sheena A. Josselyn**[2,5,6,7], **Paul W. Frankland**[2,5,6,7]*, **John Parkinson**[2,3,8‡]*

**1** Program in Molecular Medicine, Hospital for Sick Children, Toronto, Canada, **2** Program in Neurosciences & Mental Health, Hospital for Sick Children, Toronto, Canada, **3** Dept. of Biochemistry, University of Toronto, Toronto, Canada, **4** Dept. of Anatomy, Physiology and Pharmacology, University of Saskatchewan, Saskatoon, Canada, **5** Dept. of Physiology, University of Toronto, Toronto, Canada, **6** Institute of Medical Sciences, University of Toronto, Toronto, Canada, **7** Dept. of Psychology, University of Toronto, Toronto, Canada, **8** Dept. of Molecular Genetics, University of Toronto, Toronto, Canada

☉ These authors contributed equally to this work.
¤ Current address: Department of Cell Biology and Anatomy, Hotchkiss Brain Institute, Cumming School of Medicine, University of Calgary, Calgary, Alberta, Canada
‡ Lead contact.
* paul.frankland@utoronto.ca (PF); john.parkinson@utoronto.ca (JP)

**Data Availability Statement:** The sequencing data associated with this study have been deposited with links to BioProject accession number

## Abstract

During chronic infection, the single celled parasite, *Toxoplasma gondii*, can migrate to the brain where it has been associated with altered dopamine function and the capacity to modulate host behavior, increasing risk of neurocognitive disorders. Here we explore alterations in dopamine-related behavior in a new mouse model based on stimulant (cocaine)-induced hyperactivity. In combination with cocaine, infection resulted in heightened sensorimotor deficits and impairment in prepulse inhibition response, which are commonly disrupted in neuropsychiatric conditions. To identify molecular pathways in the brain affected by chronic *T. gondii* infection, we investigated patterns of gene expression. As expected, infection was associated with an enrichment of genes associated with general immune response pathways, that otherwise limits statistical power to identify more informative pathways. To overcome this limitation and focus on pathways of neurological relevance, we developed a novel context enrichment approach that relies on a customized ontology. Applying this approach, we identified genes that exhibited unexpected patterns of expression arising from the combination of cocaine exposure and infection. These include sets of genes which exhibited dampened response to cocaine in infected mice, suggesting a possible mechanism for some observed behaviors and a neuroprotective effect that may be advantageous to parasite persistence. This model offers a powerful new approach to dissect the molecular pathways by which *T. gondii* infection contributes to neurocognitive disorders.

PRJNA750929 in the NCBI BioProject database
(https://www.ncbi.nlm.nih.gov/bioproject/).

**Funding:** This work was supported by a "Chase an Idea" grant provided by the Centre for Brain and Mental Health, Hospital for Sick Children to JE, JP and PWF. Additional funding was provided by the Canadian Institute for Health Research (PJT - 152921 to JP) and a Natural Sciences and Engineering Research Council of Canada award to PWF (RGPIN-2015-03923). The funders had no role in study design, data collection and analysis, decision to publish, or preparation of the manuscript.

**Competing interests:** The authors have declared that no competing interests exist.

## Author summary

Infecting 1 in 3 worldwide, the single-celled parasite *T. gondii* causes a chronic infection whereby it can migrate to the brain and promote low-grade neuroinflammation. Given this intimate association, infection with *T. gondii* has the capacity to induce changes in brain morphology and behavior as well as modulate the production of neurotransmitters, such as dopamine. Consequently, *T. gondii* has been linked with several neurocognitive disorders including schizophrenia, dementia, and Parkinson's disease, in addition to a loss of fear response. To examine how infection impacts pathways utilizing neurotransmitters, we used a mouse model, based on stimulant-induced (cocaine) hyperactivity. Infection with *T. gondii* did not alter fear behavior but did impact motor activity and neuropsychiatric-related behaviors. Gene expression analysis revealed an initial enrichment of uninformative immune-response pathways. By applying a novel context enrichment approach, we identified significant associations with neurologically relevant genes involved in multiple pathways. These include sets of genes which exhibited dampened response to cocaine exposure in the context of *T. gondii* infection, suggestive of a neuroprotective effect that may be advantageous to parasite persistence. We suggest this novel mouse model offers a new perspective to dissect the molecular pathways by which *T. gondii* infection contributes to neuropsychiatric disorders.

## Introduction

*Toxoplasma gondii* is an obligate intracellular protozoan of the phylum Apicomplexa and is thought to infect approximately one third of the global population [1]. While the definitive hosts of *T. gondii* are felines, *the parasite* is thought to be capable of infecting any nucleated cell of any warm-blooded animal ("intermediate hosts"). During acute infection, triggered by the hosts immune response, the fast growing multiplicative form of the parasite (tachyzoite) transforms into a slower growing form (bradyzoite) that forms lifelong cysts within the host brain and muscle tissues [2]. While this chronic phase of infection is typically considered asymptomatic, its intimate association with the brain is thought to result in low-grade neuroinflammation with the capacity to induce changes in brain morphology and behavior [3]. For example, animal studies have shown that chronic *T. gondii* infection is associated with hyperactivity as well as reduced fear or increased risky behavior [4–6]. While initial studies focused on mouse hosts overcoming their aversion to cat urine [5,7], a more recent study suggests that *T. gondii* infection is associated with a more general loss of predator avoidance behavior, an important distinction that may help facilitate transmission between intermediate hosts [8]. Similar altered patterns of behavior have been observed in our closest relatives chimpanzees which are prey for leopards [9] suggesting that altered behavior patterns in *T. gondii* infected humans (which includes among others, altered response to cat urine odor) have a common evolutionary basis [10–12].

 *T. gondii* has been linked with several neurocognitive disorders including dementia, schizophrenia (SZ), and Parkinson's disease [13,14]. *T. gondii*, for example, is among the most common non-genetic risk factors for SZ, which is thought to affect ~1% of the population [15], with meta-analyses of several large scale population studies reporting an adjusted odds ratio of 1.43 [16,17]. Interestingly, in AIDS patients, immune suppression in the context of *T. gondii* infection can result in cyst reactivation and active brain cell lysis. Such effects may contribute, at least in part, to psychiatric disturbances such as delusions, auditory hallucinations and thought disorders in as many as 60% of patients [18]. Conversely, treating SZ individuals

  *Toxoplasma*-induced changes in host gene expression and behavior

seropositive for *T. gondii* with anti-parasitic drugs improves therapeutic response [19]. However, the specific pathways through which *T. gondii* may act to elicit or worsen symptoms of neurodegenerative diseases remain unknown. Potential mechanisms include changes in hormone levels [7,20], neurophysiological changes induced by neuroinflammation [21] and modulation of neurotransmitters, such as dopamine [22,23]. It is known that chronic infection with *T. gondii* results in sustained inflammation and disruption of the blood brain barrier, facilitating entry of circulating cytokines into brain tissue [24]. Further, cytokines such as interferon-gamma (IFN-γ), interleukin-6 (IL-6), IL-12b, and tumor necrosis factor (TNF) are considered signatures of *T. gondii* infection in the CNS of mice [25] and have been positively correlated with cyst load and behavior [8]. In the neuroinflammatory model, this leakage of cytokines into brain tissue results in the recruitment of immune cells leading to long-term neurophysiological changes like those observed in schizophrenia and other disorders such as autism, and bipolar disorder [3,8,26]. Interestingly, it has also been shown that cocaine activates striatal microglia, increasing production of TNF-α [27] and indeed, cocaine has been shown to induce SZ-like symptoms in mice [28]. Focusing on neurotransmission, chronic infection by *T. gondii* in a mouse model of Huntington's has been shown to activate indoleamine-2,3-dioxygenase (IDO) which catalyzes the oxidation of tryptophan, a precursor of serotonin, to produce kynurenic acid (KYNA) [23]. Reduced levels of tryptophan and an increase in neuroactive kynurenine pathway metabolites has been confirmed in *T. gondii* infected mice [29]. This attempt by the host to starve the parasite of tryptophan results in increasing oxidative stress and apoptosis. As well, KYNA can act as an antagonist of the α7 nicotinic acetylcholine receptor and may serve to modulate levels of GABA and glutamate with downstream consequences for glutamatergic, GABAergic and dopaminergic dysfunction [30]. For example, evidence suggests that GABAergic protein mislocalization via its effect on GABA signaling is responsible for seizures in mice infected with a *T. gondii* type II strain (ME49) [31]. Notably, human toxoplasmosis patients suffer from seizures and KYNA levels are elevated in patients with SZ [32].

To better understand how *T. gondii* infection contributes to neuropsychiatric disorders, we propose a novel mouse model combining *T. gondii* infection and cocaine exposure, capable of providing mechanistic insights into alterations in dopamine-related behaviors and disease pathogenesis. The basis for this model is the observation that behavioral sensitization occurs in schizophrenia and can be induced in mice by administering stimulant drugs. We compare the effects of *T. gondii* infected mice with cocaine-treated mice to investigate whether infected mice exhibit altered sensitivity to manipulations of dopamine using established behavioral paradigms (e.g., stimulant-induced hyperactivity). Using cocaine-treated mice, we examine both behavioral (stimulant sensitization, sensory motor gating and motor function) and functional consequences (differential gene expression (RNASeq) and gene set enrichment) of *T. gondii* infection in mice, to identify pathways that may be disrupted in neurological disorders.

## Results

### Cocaine treatment reveals *T. gondii* alters dopaminergic sensitivity and behaviors associated with neurological disorders but does not alter fear behavior

To examine the effects of chronic *T. gondii* infection on behaviors that are commonly impaired in neuropsychiatric disorders, we infected male mice with *T. gondii* and, following an eight-week recovery period, subjected them to a series of behavioral tests (**Fig 1A**; see Methods). For cocaine sensitization experiments, mice were injected with cocaine daily for five days five minutes before testing. One week after the end of the sensitization the mice were tested drug free

A

## Study Design

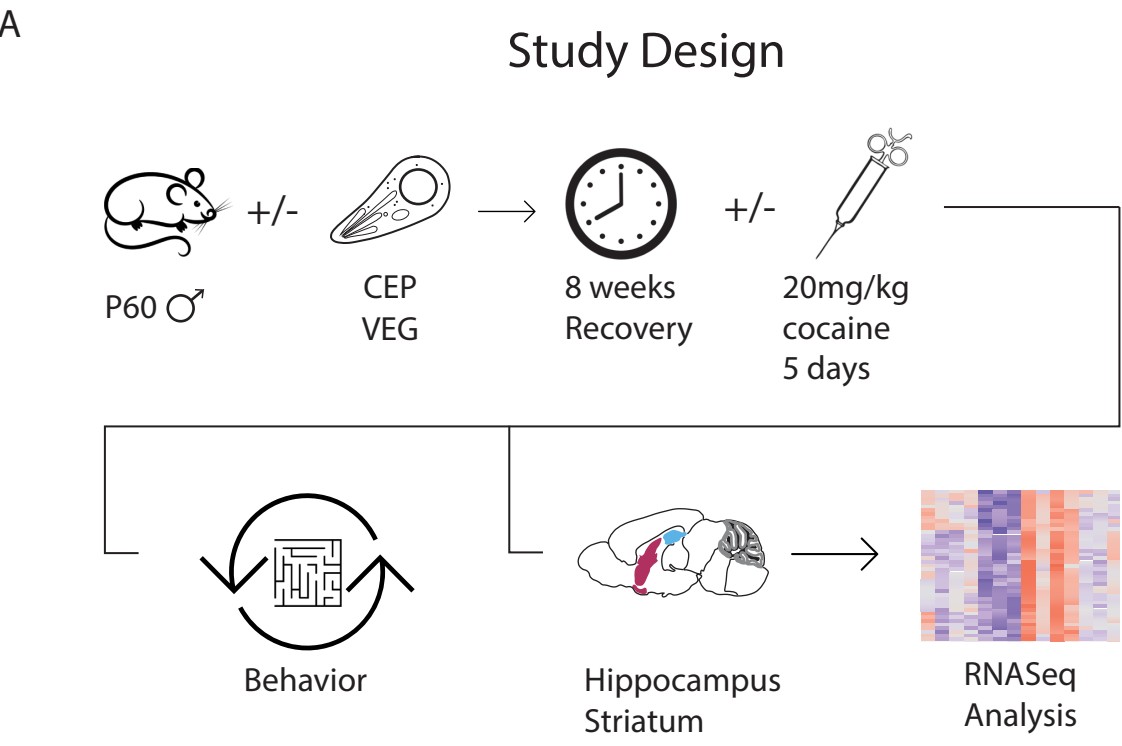

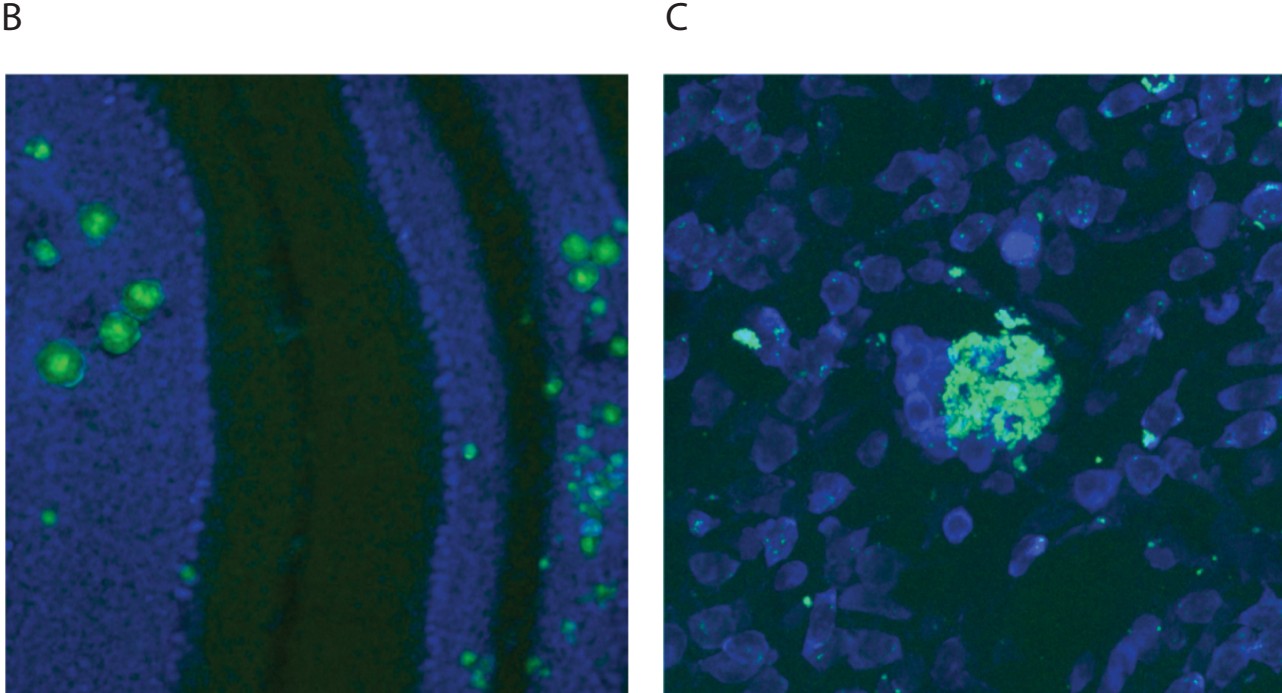

**Fig 1. Study design and histological images of *T. gondii* infection in the mouse brain.** A. Schematic overview of study design. B-C. Images showing sections of mouse brain exhibiting wide-spread infection of GFP-expressing *T. gondii* (strain CEP; green Propidium iodide counterstain; blue; larger cysts and smaller inclusions can be seen throughout the brain including in the cerebellum (B) and striatum (C).

followed by a cocaine challenge 24 hours later. The following tests were conducted: 1) fear and anxiety behaviors (contextual fear conditioning, elevated plus maze (EPM) and open field); 2) motor ability (rotor rod); and 3) neuropsychiatric-related behaviors (cocaine sensitization and prepulse inhibition). Prepulse inhibition (PPI) is a measure of sensory-motor gating that evaluates ability to filter out irrelevant stimuli. In this case, this was assessed using the acoustic startle response. We used this task because it is impaired in (i) schizophrenia patients and (ii) rodent models of schizophrenia, and (iii) impairments in rodent models are reversed with antipsychotic drugs. To assess strain specific effects, mice were infected with one of two different type III strains of *T. gondii*: CEP and VEG. Strains were chosen due to their lower virulence in mice and ability to form brain cysts. As expected, histological examination confirmed the presence of cysts in the brains of infected mice (**Fig 1B**). Since both strains of *T. gondii* produced largely similar results (**S1 Fig**), here we report the results only for strain CEP. An additional cohort of mice was used to harvest striatum and hippocampus samples for histology and RNA sequencing.

We found that mice infected with *T. gondii* acquired contextual fear conditioning normally. There were no differences in freezing behavior over the 5 minute duration (Repeated measures ANOVA effect of group $F_{(1,37)} = 0.048$, $p = 0.83$; **Fig 2A**) or freezing behavior during training (Minute by group: $F_{(4,148)} = 0.076$, $p = 0.99$; **Fig 2B**), indicating that shock reactivity is unaltered in infected mice. *T. gondii* infection also did not alter the subsequent retention of the contextual fear memory (independent samples t-test, $t_{(39)} = 0.91$, $p = 0.37$) or anxiety-associated behaviors. Infected mice did not show altered anxiety-related behavior in the elevated plus maze (**Fig 2C**). They did not spend significantly more time than control mice in the open arms (Two-way ANOVA, Zone x Group: $F_{(1,37)} = 0.067$, $p = 0.80$) and there was no significant difference in the distance travelled during the elevated plus maze test ($t_{(37)}$– 1.68, $p = 0.11$; **Fig 2D**). Further, we observed no significant group differences in the time spent in each zone of the open field test (Two-way ANOVA, Group × Zone: $F_{(2,74)} = 1.67$, $p = 0.20$; **Fig 2E**). We found that infection with *T. gondii* significantly reduced overall distance travelled in the open field test ($t_{(37)}$– 2.99, $p = 0.0049$; **Fig 2F**).

To assess whether dopaminergic sensitivity is altered in infected mice, we examined behavioral sensitization to the hyper-locomotion inducing effects of cocaine (**Fig 2G–2I**). Mice were first habituated to the test arena over three sessions. There was a significant difference in locomotor activity between groups (Main effect of group $F_{(1,99)} = 4.70$, $p = 0.0001$; **Fig 2G**) driven by the first session but no difference between groups by the end of habituation. We found that *T. gondii* significantly altered behavioral response to cocaine (Three-way ANOVA, significant group by drug by session effect: $F_{(4,120)} = 4.13$, $p = 0.0036$; **Fig 2H**). Specifically, infected mice show an equivalent initial locomotor response to cocaine (day 1, *p = 0.97*) but show a significantly blunted sensitization upon repeated cocaine administration (Post-hoc Newman-Keuls test; Days 4 and 5, *p ≤ 0.0011*). We also observed a lasting effect of infection on locomotor activity when mice were tested one week following sensitization (Two-way ANOVA main effect of group $F_{3,60)} = 9.39$, *p = 0.000035*; **Fig 2I**). Control and infected mice administered saline (not cocaine) show comparable locomotor activity (Post-hoc Newman-Keuls test *p = 0.63*) when tested one week later. But, infected mice that had *previously* been exposed to cocaine showed a blunted response compared to control mice challenged with cocaine (Post-hoc Newman-Keuls test, *p = 0.030*) although their locomotor response did not differ significantly from mice that had not previously received cocaine (Post-hoc Newman-Keuls test, *p = 0.061*).

In terms of motor activity, we additionally found that *T. gondii* infection slowed rotarod acquisition (Repeated measures ANOVA, significant session x group interaction, $F_{(4,37)} = 4.00$, $p = 0.0041$). Specifically, control mice outperformed *T. gondii* infected mice on the final

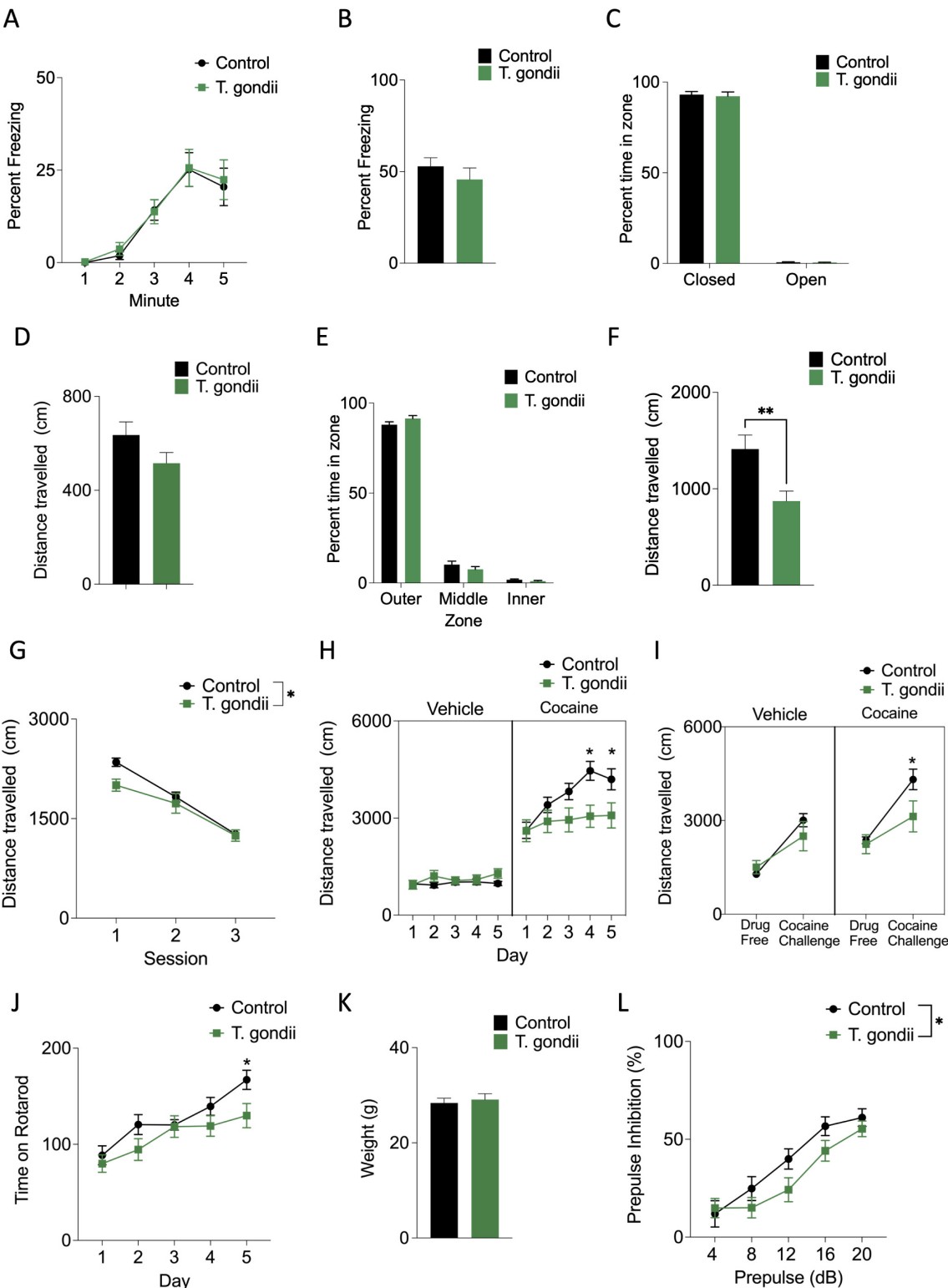

**Fig 2. Behavioral characterization of *T. gondii* (CEP) injected mice.** A-B. Infected mice exhibited normal acquisition (A) and expression (B) of contextual fear memory. C-D. Anxiety related behaviors including time spent in the open arm of the elevated plus maze (C) or distance travelled (D) did not differ between infected and control mice. E. There was no difference in the anxiety related measure of time spent in the inner zone of the open field. F. Infected mice travelled significantly less distance in the open field compared to controls. G-I. In the behavioral sensitization paradigm mice were first habituated to the testing arena which eliminated pre-existing

differences in locomotor activity (G). *T. gondii* infected mice showed a blunted sensitization to the effects of cocaine, as exhibited by a significantly decrease in distance travelled following successive administrations of cocaine (H). One week following sensitization *T. gondii* infected mice still exhibited a blunted response to a challenge dose of cocaine compared to control mice (I). J. On the rotarod, *T. gondii* infected mice exhibited impaired acquisition relative to control mice indicative of potential motor impairments. K. There was no significant difference in the weight of *T.gondii* and control mice. L. *T. gondii* infected mice exhibited blunted prepulse inhibition compared to control mice.

day of training (*p = 0.033)* (Fig 2J). These results are consistent with previous studies that have shown *T. gondii* infection results in motor deficits in mice without major brain damage or cognitive dysfunction [33]. There were no differences in weight between *T. gondii* infected and saline treated mice to account for the differences in rotarod performance (t(37) = 0.46, *p =* 0.57; Fig 2K). It is possible that locomotor differences could be driven in part by sickness behaviours induced by *T. gondii* infection. While reduced locomotion in the open field is consistent with such an account, we noted no overt signs of ill-health in *T. gondii*-infected mice (e.g., weight loss, lethargy etc). Moreover, normal performance in other behavioural-tests (contextual fear conditioning, elevated plus maze) is generally inconsistent with a sickness based account.

We also examined paired-pulse inhibition, a behavior that is commonly disrupted in neuropsychiatric conditions, as well as in their animal models. Here we observed that infected mice exhibited an impaired response compared to control mice (Fig 2L). Levels of PPI increased with prepulse intensity (Two-way ANOVA, main effect of prepulse intensity F(4,110) = 6.69, *p = 0.000074*). However, this increase was blunted in infected mice (Two-way ANOVA, main effect of group F(1,110) = 4.79, *p = 0.031*).

While infected mice did not exhibit altered fear or anxiety related behaviors, we observed alterations in motor performance and interestingly in behaviors that are commonly associated with neuropsychiatric-conditions. These observations are significant given the previously described link between *T. gondii* infection and risk of neurocognitive disorders such as schizophrenia. Notably, these effects were largely replicated for two different strains of the parasite (S1 Fig). One exception was PPI. While strain VEG also elicited changes in sensory-motor processing relative to control mice, the response appeared enhanced. In the following sections we investigate how changes in gene expression, associated with chronic infection by *T. gondii*, may contribute to these changes in behavior. Since behavioral differences were largely consistent for both strain VEG and CEP, only a single strain (CEP) was used for expression profiling.

## Exposure to cocaine and chronic infection by *T. gondii* result in distinct, tissue-specific, and treatment-specific gene expression profiles

To examine the impact of cocaine and *T. gondii* infection on gene expression in the mouse brain we performed RNASeq on the striatum and hippocampus from samples collected from all four treatment groups (+/- cocaine and +/- *T. gondii*). Both tissues are established sites of *T. gondii* infection [34,35]; striatum was selected for its role in motor functions and response to rewarding and aversive stimuli [36], while the hippocampus is involved in memory and learning [37]. Four biological replicates were performed. The resulting ~50M reads per sample were filtered for quality and mapped to the mouse genome using STAR [38], resulting in a matrix of read counts for each gene/sample. Mouse gene counts were normalized and analyzed using DESeq2 (see Methods–Statistical Analysis). Out of a possible 8,563 *T. gondii* genes only 351 were detected and only 3 with greater than 5 reads, consistent with the relative abundance of mouse tissue and lower activity associated with the bradyzoite stage of the parasite [39]. Over all conditions, we identified 4439 genes that were differentially expressed (DE) relative to controls (S1–S3 Data). These include 2912 genes associated with the hippocampus samples and

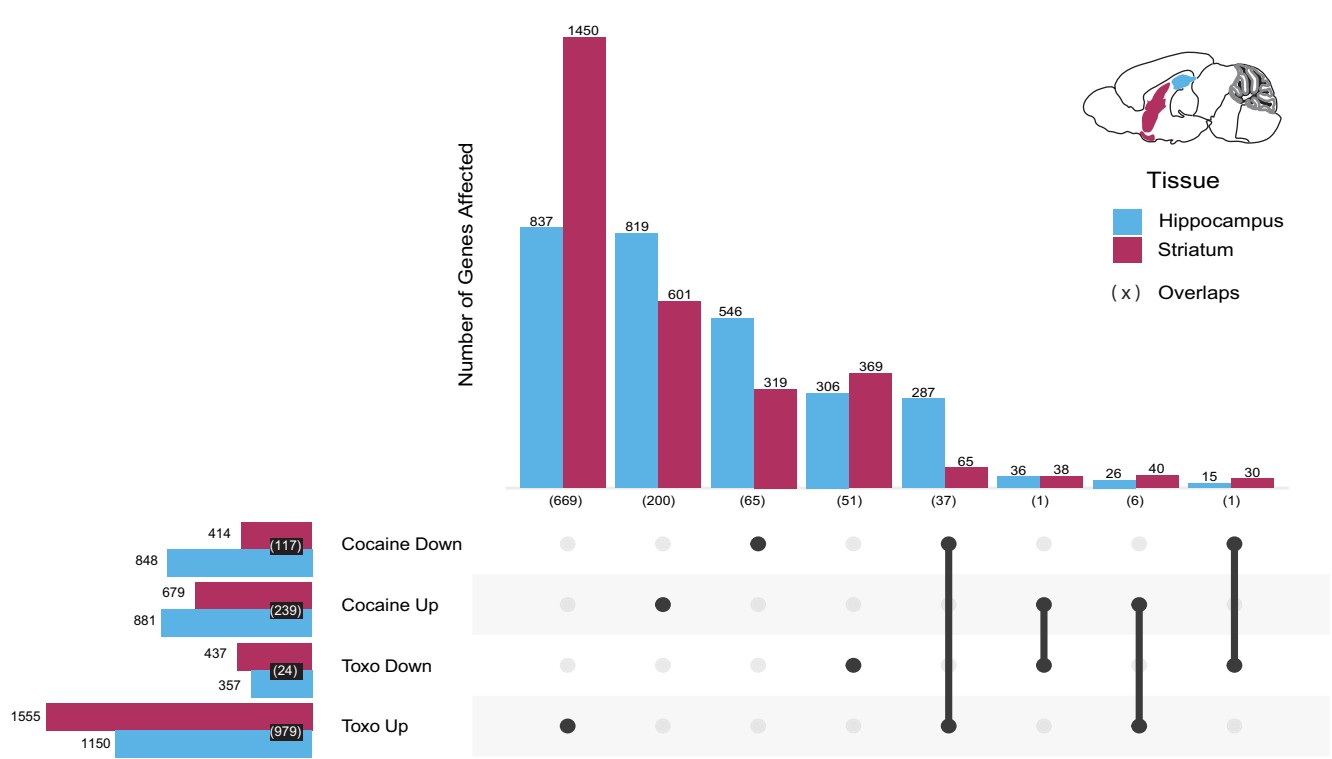

**Fig 3. Expression of differentially expressed genes responding to cocaine and *T. gondii* infection.** The upset plot shows the number of differentially expressed genes by response, with overlaps by treatment category and tissue. The horizontal bars to the left indicate the total number of differentially expressed genes associated with each of the four conditions. The filled in dots in the dot matrix, together with the corresponding vertical bars then show how these genes are distributed. For example, of the 1150 differentially expressed genes in the Hippocampus associated with "Toxo Up", 837 are only associated with the "Toxo Up" dataset (1st column), while 287 are shared with the "Cocaine Down" dataset (5th column) and 26 are shared with the "Cocaine Up" dataset (7th column). Within categories, tissue overlaps are indicated (bracketed). Genes represented are mouse genes identified as being significantly differentially expressed (likelihood ratio test, FDR<0.1) across pairs of conditions within a tissue.

3077 genes associated with the striatum samples. 1550 of these genes were common to both tissues.

A majority of DE genes (2344 and 2739 for hippocampus and striatum respectively) were associated with a single treatment (either *T. gondii* infection or cocaine), and these treatments resulted in more genes being upregulated than downregulated (**Fig 3**). Furthermore, across most of the single-treatment outcomes (i.e., up or downregulated upon treatment with cocaine or *T. gondii*), gene expression changes appeared to be tissue specific with the number of DE genes common to both (3.0–15.3%) significantly smaller than expected (25.9%; Fisher's Exact Test p = 3.73e-12 to p = 2.20e-16; **S4 Data**). DE genes exhibiting upregulation in response to *T. gondii*, comprising the largest gene set for both tissues (hippocampus = 1150; striatum = 1555) were an exception, showing greater overlap than expected (36.2% overlap versus an expected 25.9%, Fisher's Exact Test p = 2.20e-16). Notably, a larger number of genes responded to the cocaine treatment in the hippocampus (Fisher's Exact Test p = 7.35e-15), while more genes responded to *T. gondii* in the striatum (Fisher's Exact Test p = 2.05e-14), likely reflecting the different targets of the two treatments. Among the genes responding to both treatments, the largest overlap between the two tissues were 37 genes that were significantly upregulated in response to *T. gondii* infection, but downregulated in response to cocaine, suggesting differing effects on the same targets. While cocaine-responsive and

toxoplasma-responsive gene expression changes have previously been reported independently [40–42], to our knowledge a direct comparison of hippocampal and striatal gene expression changes in response to these factors and in combination has never been done. In the next section we attempt to identify the functional consequences associated with these changes in gene expression.

## Gene set enrichment identifies a wealth of neurologically relevant GO biological processes statistically enriched in DE genes

To identify pathways and processes impacted by exposure of the mouse brain to *T. gondii* and/ or cocaine we performed gene set enrichment analysis (GSEA) using the Cytoscape plug-in, BiNGO [43]. This initial analysis confirmed a robust, general immune response to *T. gondii* infection (**S5 Data**). However, the large number of enriched terms (2755 and 2014 for hippocampus and striatum respectively) render it challenging to identify consistent functional themes whether using the full gene set or, subsets representing each of the eight categories of response. Similar observations were obtained using the largely independent Reactome and KEGG pathway data sets (see Methods and **S6 Data**).

Given our interest in the neurological significance of DE genes and how these might vary across the treatments and tissues, we therefore developed a literature-guided, systematic approach to filter neurological-related terms from the larger sets of 2000+ terms identified as significantly enriched (see Methods). This process allowed us to collapse 734 GO terms into 34 distinct higher-level 'term groups' (**S7 Data**). Each of these 34 term groups were then assigned to a treatment/tissue if at least one GO term associated with that group was enriched in that treatment/tissue (**Fig 4A and 4B**). Among the genes upregulated by *T. gondii* were many involved in neuroinflammation processes that are typically only expressed during infection. Neuroinflammation has been linked to several neuropsychiatric disorders (e.g., Parkinson's disease) which also correlate with *T. gondii* infection [13] and it has been suggested that this may be driven through the activation of microglia and astrocytes in response to latent toxoplasmosis [44,45].

Among the DE genes we identified, 71 were associated with the genesis, growth, activation, or regulation of glial cells, including 19 associated with microglia and 15 associated with astrocytes (**S7 Data**). Our model confirms upregulation of TNF-alpha, a protective inflammatory cytokine released by activated microglia, in response to *T. gondii* infection [46]. It is particularly interesting that TNF-alpha can result in internalization of synaptic AMPA receptors on striatal neurons, depressing cocaine-induced behavioral sensitization [27]. Normally, repeated doses of cocaine are needed to reduce sensitization. However, the persistent activation of microglia by *T. gondii*, would explain the observed lack of sensitization in these mice upon cocaine challenge. A further 64 DE genes upregulated by *T. gondii* were associated with the growth and proliferation of dendritic cells which have been shown to play key roles in host resistance to *T. gondii* [47] as well as contributing to various forms of neuroinflammation associated with neurodegenerative autoimmune disease, injury and CNS infections [48]. Among genes that are upregulated in response to *T. gondii* and cocaine we identified 247 genes associated with mitogen-activated protein kinases (MAPKs) as well as 105 DE genes annotated to NFκB signaling, regulation and activation. MAPKs are crucial regulators in the production of inflammatory mediators [49] whereas NFκB is one of the most important transcription factors for many proinflammatory genes.

In addition to eliciting responses related to neuroinflammation, we also identified 73 DE genes responding to cocaine and/or *T. gondii* associated with terms relating to the metabolism, transport, secretion and regulation of various neurotransmitters and their receptors. Further,

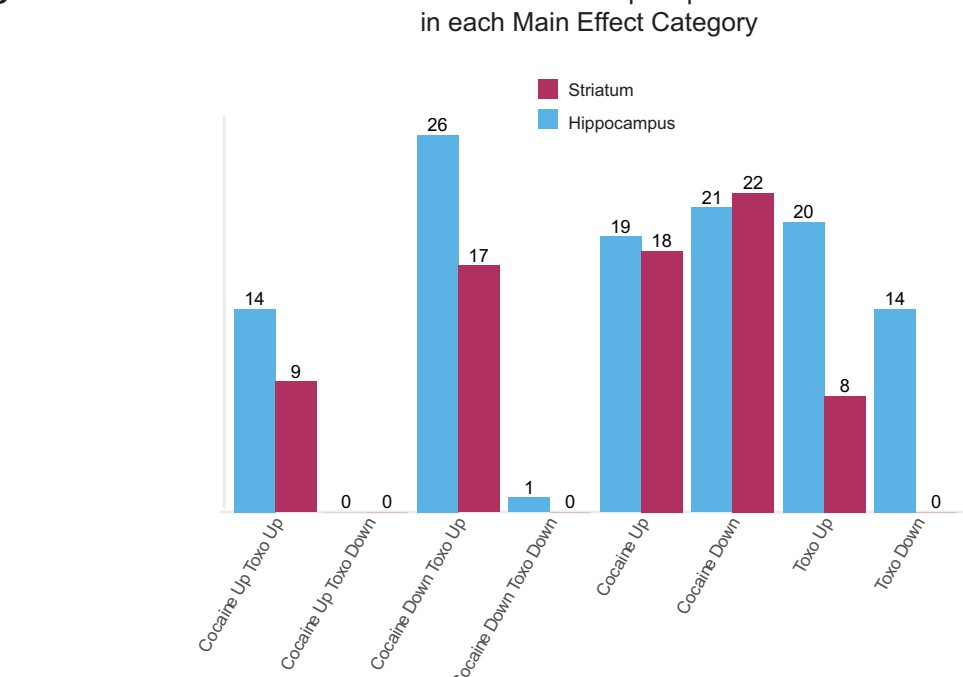

**Fig 4. Summary of neurologically relevant, enriched annotations.** A. Occurrence of neurologically relevant term groups across main effect and interacting effect categories in hippocampus and striatum. B. Number of term groups represented in each main effect category in each tissue.

185 DE genes relate to synapses and synaptic transmission. Interestingly, we observed significant down-regulation of two positive regulators of calcineurin signaling (LMCD1 and SLC9A1) in response to cocaine and *T. gondii* in hippocampus. Calcineurin has been linked to receptors for several brain chemicals including dopamine, glutamate and GABA [50]. Calcineurin deficient mice have been reported to exhibit symptoms similar to humans with schizophrenia including working memory impairment, attention deficits and, impaired social behavior [51] and, a reduction in calcineurin expression in the hippocampus of (human) SZ patients has been reported [52,53]. Given previous associations between *T. gondii* infection and KYNA production [23], and its link to SZ [45], we were intrigued to find that in response to *T. gondii* infection kynureninase (KYNU) [EC:3.7.1.3] was significantly upregulated in both tissues, while the adjacent enzyme in the pathway, 3-hydroxyanthranilate 3,4-dioxygenase (HAAO) [EC:1.13.11.6] was also upregulated in the striatum; cocaine exposure had no impact.

Despite the large number of GO-terms identified as enriched in DE genes, by collapsing multiple neurological-related terms into 34 term groups, we were able to identify a wealth of genes implicated in modulating the host brains response to infection by *T. gondii*. Nevertheless, neurologically relevant terms ranked among the least significantly enriched GO annotations. To address this, in the next section we show how these analyses can be further enriched through developing a customized ontology.

## Recovery of neurologically relevant annotations is aided by a novel, context enrichment approach

We propose that DE genes identified in this study, represent two distinct subsets, one associated with a robust, highly enriched, general immune response and the other, representing a smaller subset associated with neurological effects that is masked by a loss of statistical power associated with multiple testing correction. To test this, we first defined a set of root keywords ("neuro", "brain", "behavior", "locomot", "memory", and "synap") which we considered, based on our curation of the literature, to be both relevant to the context of our study and non-immune system specific. An AmiGO search [54] for GO biological process terms associated with these keywords was then performed and resulted in 1632 GO terms which we deemed collectively to define the functional "context". The terms retrieved by each keyword (**S8 Data**) were largely complimentary as shown by the relatively small number of overlapping terms (**S2 Fig**). Using the collated full-text descriptions of these GO terms, an analysis of word frequencies confirms that they represent a full range of neurologically relevant concepts (**Fig 5A**). These terms were next used to filter the DE gene set, removing genes that were not associated with at least one annotation matching the defined context, a process we refer to as "context enrichment". The intent was to distill, from among 4439 DE genes identified across both tissues, a neurologically related DE subset distinct from those genes relating to the expected, general immune response.

GSEA was performed on the resulting 641 context-enriched genes (466/2912 hippocampus and 450/3077 striatum genes, with an overlap of 275 genes) and results were visualized for each tissue as a Cytoscape network (**S9 Data**). Improving our focus on neurologically related genes, these graphs capture many of the processes identified in our earlier curation approach and again highlight the diverse range of biological processes potentially contributing to neurological effects. Comparing the position of the top 20 enriched GO annotations in the context-enriched set relative to the same annotations in the full set of GO terms, context-enriched terms ranked considerably higher, with more significant corrected p-values than in the full set (**Fig 5B**). For example, the top ranked term in the GSEA results for hippocampus context-enriched genes, "Locomotory Behavior" was the 321[st] term result for DE genes. Further, the

A

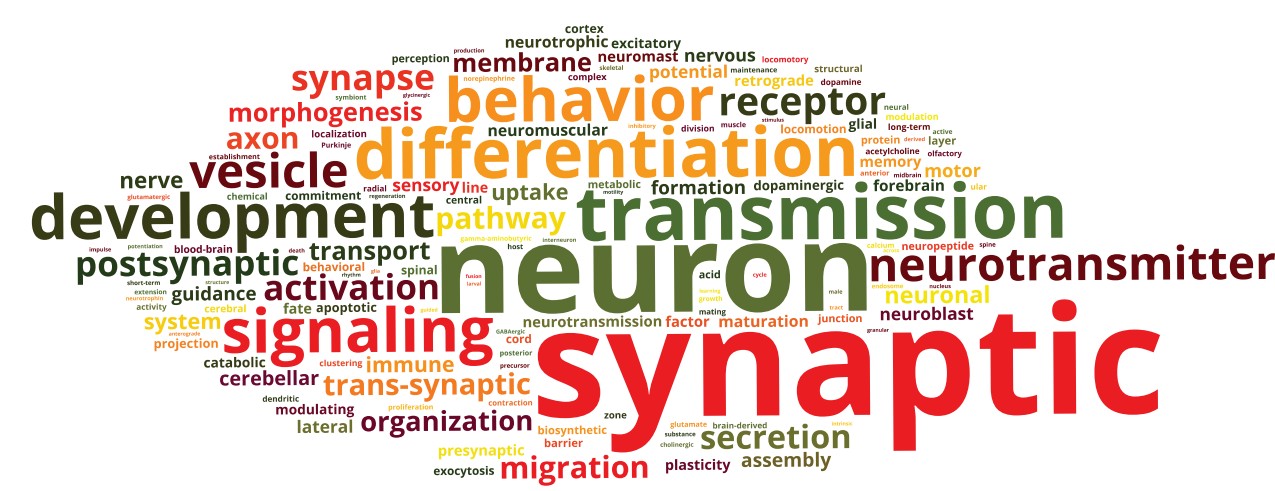

B

## Top 20 Enriched Annotations

### Hippocampus

| NIGO | GO | DE Rank | Term |
|---|---|---|---|
| | | 96 | locomotory behavior |
| | | 167 | positive regulation of locomotion |
| | | 223 | locomotion |
| | | 321 | nervous system development |
| | | 375 | behavior |
| | | 414 | generation of neurons |
| | | 435 | brain development |
| | | 502 | regulation of nervous system development |
| | | 517 | neurogenesis |
| | | 678 | central nervous system development |
| | | 684 | neuron differentiation |
| | | 728 | regulation of neuron differentiation |
| | | 793 | neuron development |
| | | 974 | regulation of neurogenesis |
| | | 975 | neuron projection development |
| | | 1006 | regulation of locomotion |
| | | 1196 | neuron projection morphogenesis |
| | | 2195 | regulation of synaptic transmission |
| | | - | transmission of nerve impulse |
| | | - | neurological system process |
| | | - | regulation of neurological system process |
| | | - | neurite development |
| | | - | axonogenesis |
| | | - | regulation of transmission of nerve impulse |
| | | - | synaptic transmission |

Significance (FDR) — Most significant → Least significant

### Striatum

| NIGO | GO | DE Rank | Term |
|---|---|---|---|
| | | 40 | regulation of response to external stimulus |
| | | 89 | positive regulation of response to external stimulus |
| | | 163 | regulation of locomotion |
| | | 228 | positive regulation of locomotion |
| | | 377 | locomotion |
| | | 707 | nervous system development |
| | | 852 | neurogenesis |
| | | 959 | generation of neurons |
| | | 987 | regulation of nervous system development |
| | | 1031 | regulation of neurogenesis |
| | | 1221 | ensheathment of neurons |
| | | 1222 | axon ensheathment |
| | | 1803 | regulation of neuron differentiation |
| | | - | behavior |
| | | - | neuron differentiation |
| | | - | neuron development |
| | | - | locomotory behavior |
| | | - | neurological system process |
| | | - | central nervous system development |
| | | - | neuron projection development |
| | | - | transmission of nerve impulse |
| | | - | brain development |
| | | - | neurite development |
| | | - | learning or memory |

Significance (FDR) — Most significant → Least significant

Legend:
Hip / Str — Complete representation, Partial representation (GO / NIGO)
● Term represented
○ Term not represented
●—● Term identity

**Fig 5. Context enrichment of DE genes.** A. Word cloud representation of word frequencies in a list of full-text descriptions of GO terms defining the context filter. These GO terms result from an AmiGO search based on six key words defining a neurological 'context'. Here the word cloud indicates the frequency of words in the list, with more frequent terms illustrated as larger words. B. For the context filtered subset of DE genes, a comparison of top 20 enriched annotations (GO vs. NIGO) in hippocampus and striatum and their relative rankings in a GO enrichment analysis of the full DE gene set. The significance of each set of 20 terms for the genes associated with the hippocampus ranged from $1.21E^{-87}$–$2.89E^{-27}$, $2.4E^{-98}$–$3.69E^{-25}$ and $4.62E^{-31}$–$3.69E^{-2}$ for the NIGO, GO and DE Rank respectively. The significance of each set of terms for the genes associated with the striatum ranged from $8.14E^{-67}$–$4.1E^{-16}$, $1.19E^{-72}$–$1.52E^{-17}$ and $3.83E^{-54}$–$3.05E^{-2}$ for the NIGO, GO and DE Rank respectively.

relative order of these top twenty terms was also scrambled between gene sets while among the top 20 terms, 7/20 in hippocampus and 11/20 in striatum were not detected as significant in the larger GO gene set. Indeed, of the 641 context-enriched genes, 114 were missed in the larger analysis illustrating how genes and terms associated with immune responses masked our ability to detect a greater depth of neurologically relevant signals. By way of example, we discovered a modest but significant overlap (Fisher's Exact Test p = 0.0492) between our context-enriched gene set and genes highlighted in a recent meta-analysis of differentially methylated positions associated with psychosis, schizophrenia and treatment-resistant schizophrenia [55] which was not detectable using the full DE gene set.

To further validate our context enrichment approach, we performed GSEA using NIGO, a GO-derived subset focusing on neuro-immune function terms [56]. Comparing the top 20 term results obtained using the NIGO dataset with our enriched dataset revealed a high degree of correspondence in ranking of terms, confirming our ability to capture relevant terms (**Fig 5B**). While NIGO might appear to represent a suitable solution to the enrichment analysis we note that over the past decade since NIGO was published, considerable efforts have been made to address deficiencies in GO with respect to both neurologically relevant gene annotations and the ontology itself [57–59]. Thus, we expect many terms to be present among current DE gene annotations that are not reflected in NIGO and therefore excluded from the analysis. For example, terms such as 'neuron projection development' and, 'neuron projection morphogenesis' among the top 20 in our context-enrichment analysis were missing from NIGO annotations as denoted by the open circles in **Fig 5B**. Such missing terms likely account for observed discrepancies in term detection and ranks between NIGO and GO for our AmiGO-based set. Updating NIGO would require a systematic clipping of the Gene Ontology and was therefore outside the scope of our study. Nevertheless, the simpler approach developed here proved capable of overcoming current limitations of GO annotations by revealing additional neurologically relevant terms and could be readily applied more widely to focus on other specialized gene sets representing additional functional or disease contexts.

## Gene subsets present in both striatum and hippocampus exhibit complex responses to the two treatments implying they operate in parallel or intersecting pathways

Leveraging our statistical model we identified subtypes of genes responding to cocaine and parasitic infection where the change in expression was not simply additive (i.e. sum of effect of each treatment in isolation). In total we identified 40 hippocampus and 165 striatum DE genes, that could be organized into 8 'functional effect' groups based on their gene expression patterns (**S3 Fig**, **S10 Data**). To set the stage for future follow up investigations we highlight several examples of antagonistic or synergistic interactions suggestive of competitive or complementary pathways.

For both tissues, the most common type of interaction involved genes that were downregulated in response to either cocaine or *T. gondii*, but no significant change in expression in combination (i.e., rescue of the *T. gondii* and cocaine effects; subgroup 'b', **Fig 6A**). This pattern

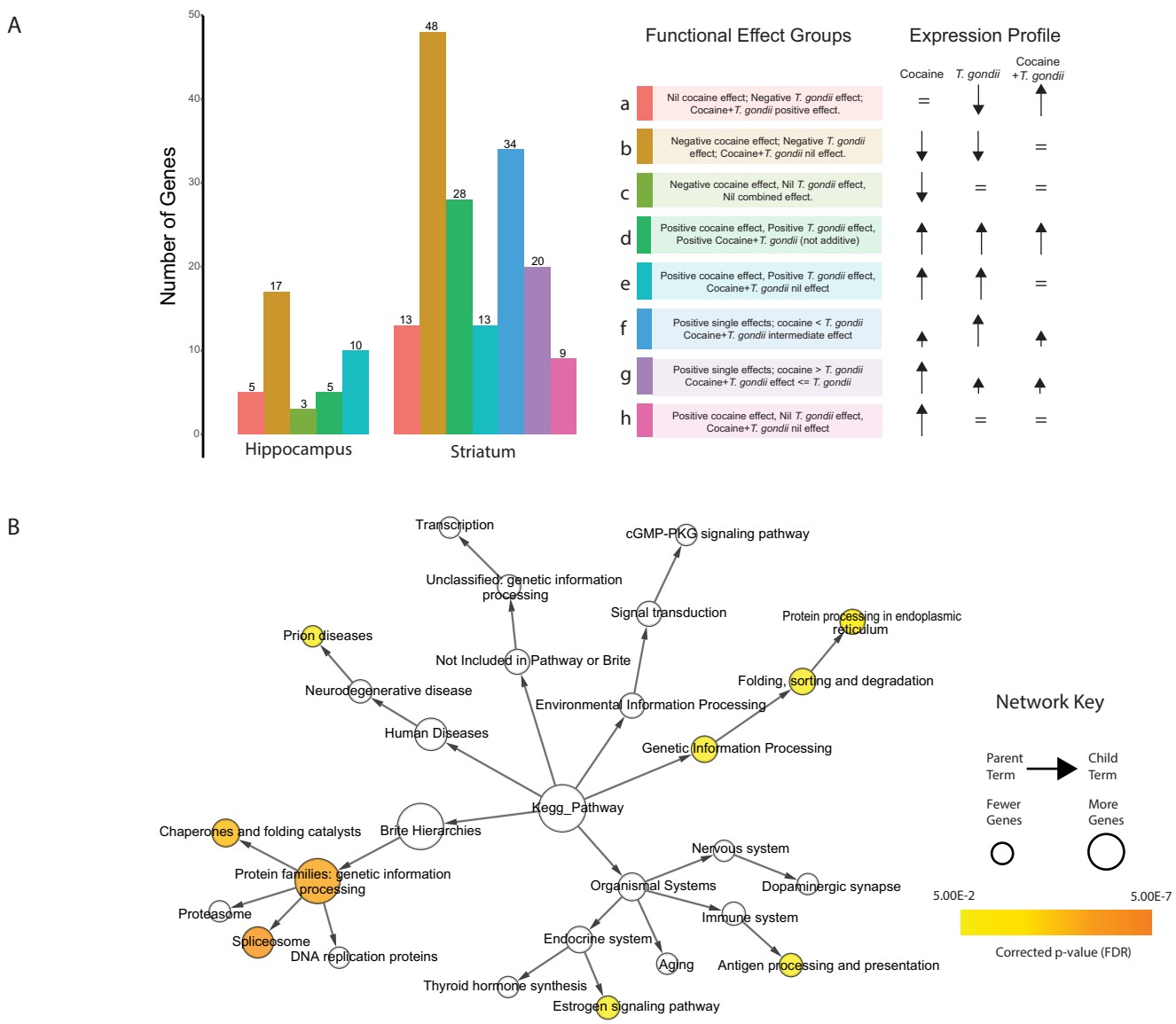

**Fig 6. Functional characterization of mouse genes with interacting effects.** A. Frequency of mouse genes with statistically significant interactions by functional effect category (hippocampus and striatum). B. Network visualization of KEGG pathway enrichments for functional effect group b in striatum (largest category) providing insight into parent-child relationships of significantly enriched terms. For example, "spliceosome" and "chaperones and folding catalysts" can be seen to be most specifically as well as most significantly enriched. For context, all significantly enriched terms before correction for multiple testing are shown. Colors represent significant terms after multiple testing correction (hypergeometric test, FDR < 0.05).

was enriched relative to other patterns (p < 0.01 for both tissues; Chi-square Goodness of Fit Test). GSEA of KEGG pathways for this subgroup suggested that cocaine may compete with or otherwise impede *T. gondii* in its attempt to down-regulate expression of host proteins critical to the fidelity of cellular protein production at a post-transcriptional and/or post-translational level. We found that 27/31 annotated striatum proteins in this set were associated to the KEGG pathway "Protein families: genetic information processing" and, significant KEGG pathway enrichments included "spliceosome" (Hypergeometric test p = 4.6596E-6) and "chaperones and folding catalysts" (Hypergeometric test p = 1.5494E-4; **Fig 6B, S5 Data**). Also included in this subgroup are two genes (*ATF2* and *ARNTL*) annotated to "Dopaminergic

synapse". Altered *ATF2* expression has been associated with progressive reduction in postsynaptic dopamine response [60] whereas ATF2 with ARNTL (and CLOCK) comprise a heterodimeric enhancer of circadian clock genes (*PER1*, *PER2*, *PER3*). The latter are indeed differentially expressed, although they are not within this subgroup. Notably, the dysregulation of circadian rhythms has been associated with neurological disorders involving dopamine including Parkinson's Disease [61].

Focusing on functional effects in the hippocampus, we identified a unique subgroup (subgroup 'c', **Fig 6A**) comprising 3 DE genes whose expression was downregulated by cocaine, with no response to *T. gondii* or the combined condition (i.e., rescue of the cocaine effect). These were *VGLL4*, encoding a transcriptional co-factor [62]; *SOX21*, a transcriptional activator [63]; and *PSTPIP2*, whose product regulates F-Actin bundling and enhances motility in macrophages [64]. *SOX21*, is associated with the GO terms "positive regulation of myelination", "axon ensheathment in central nervous system", "central nervous system myelination", and "negative regulation of Schwann cell proliferation". From a neurological perspective, it is interesting to note that a gene or genes that may cause dis-regulation of myelination in response to cocaine sensitization are mitigated or masked in the presence of *T. gondii*. Beyond the trivial explanation of *T. gondii* infection rescuing the cocaine effect, the alternative explanation is that these may represent functions that the parasite needs to ensure are not activated through other stimuli.

Turning to the striatum, we identified three unique interacting subgroups (subgroups 'f', 'g' and 'h'; **Fig 6A**). While subgroup f was functionally diverse, the other two subgroups were enriched with annotations associated with apoptosis and cell death (subgroup 'g') and TGFbeta signalling (subgroup 'h') respectively. Both latter subgroups of genes exhibited a greater response to cocaine alone than either *T. gondii* or the combination of *T. gondii* and cocaine. While TGFbeta has been associated with neuroinflammatory pathways and identified as neuroprotective following brain ischemia [65], genes in subgroup 'h' include *NNROS* and *FBN1* which are associated with negative regulation of the TGFbeta signalling pathway. We therefore speculate that *T. gondii* infection may exploit these neuroprotective pathways to allow the parasite to persist, even in the context of the cytotoxic effects elicited by cocaine exposure.

## Discussion

Given that hyperactive dopaminergic function is a hallmark of schizophrenia, we developed a mouse model with associated behavioral tests to reveal altered dopaminergic sensitivity in *T. gondii* infected male mice and focused on identifying the molecular pathways by which *T. gondii* infection contribute to disease pathogenesis. We stress that this model is not a model of schizophrenia or any other neurological condition *per se*. Rather, we are modeling a behavioral effect that occurs in schizophrenia namely altered dopamine function with concomitant changes in dopamine-related behavior (cocaine sensitization) which is relevant because *T. gondii* infection is associated with altered dopamine function and increased odds ratio for schizophrenia. Using this model, we found that infected mice exhibited no change in memory or anxiety-associated behaviors. This contrasts with a previous study based on infection with strain ME49, in which the authors reported that infected mice spent significantly more time in the open arms of the EPM and an increase in exploratory behaviors in the open field (although time spent in the central zone of the field was reduced) [8]. Such differences may arise as a consequence of the breed of mouse or strain of parasite used. Here we used offspring from a cross between C57BL/6NTacfBr [C57B6] and 129Svev [129] mice, together with *T. gondii* strains VEG and CEP. The earlier study used B6CBAF1/J mice infected with the type II *T. gondii*

strain, ME49 [8]. The type III strains, VEG and CEP exhibit lower virulence in mice relative to ME49, and we used an initial inoculum of ~500,000 parasites compared to 50–1000 parasites used in the other study. The decrease in anxiety and fear-related behaviors that have been observed in some previous studies following *T. gondii* infection are typically described as behavioral manipulations that facilitate predation mediated transmission of the parasite from intermediate to definitive hosts. However, there are other behavioral changes that may serve the same purpose such as decreased motor activity or coordination as we observed here.

Beyond memory and anxiety-associated diseases, we further found that infection with *T. gondii* significantly reduced locomotor activity levels in both the open field test and the elevated plus maze and impaired rotarod acquisition consistent with previous studies [33]. Finally, cocaine sensitization resulted in *T. gondii* infected mice showing an equivalent initial locomotor response to cocaine (day 1) but exhibited blunted sensitization upon repeated cocaine administration. Furthermore, while drug free control and infected mice show normal activity level, infected mice exhibited reduced sensitization to cocaine after one week of abstinence. Finally, our PPI experiments also revealed changes in sensorimotor processing in response to *T. gondii* infection. Notably, unlike other tests, the two strains exhibited contrasting behaviors, with strain CEP exhibiting a blunted response relative to control mice, while strain VEG exhibited an elevated response. While these different responses may indicate the sensitivity of sensorimotor processing to subtle genetic differences between the two strains, they nevertheless affect the same neuropsychiatric relevant mechanism, albeit in different directions.

In terms of behavior, our primary goal in the current study was to determine the impact of *T. gondii* infection on behaviors known to be impacted in neuropsychiatric conditions. Our findings in the cocaine sensitization and PPI experiments clearly indicate that *T. gondii* alters the normal pattern of behaviors in these tasks although, with the exception of PPI experiments involving VEG-infected mice, these effects were unexpectedly in the opposite direction of what is typically observed in individuals with schizophrenia. Interestingly, we noted that *T. gondii* infection resulted in a lack of sensitization in mice upon cocaine challenge. Such depressed sensitization may be driven through the persistent activation of microglia, resulting in the elevated expression of TNF-alpha and subsequent internalization of synaptic AMPA receptors [27]. In any case, given the magnitude of *T. gondii* infection as a risk factor for schizophrenia, the infection related behavioral sensitization to cocaine and the corresponding changes in gene expression may provide critical insight into disease etiology.

Following behavioral tests, gene expression profiling on two regions of the mouse brain, hippocampus and striatum, identified 4,439 unique genes that are significantly differentially expressed in mice in response to chronic *T. gondii* infection and/or pre-sensitization to cocaine. At least 2024 of these genes are associated with neurologically relevant functional annotations implicating a broad range of potential contributing pathways. Most expression changes we observed consisted of genes responding positively to either *T. gondii* or cocaine, suggesting that while they may have complimentary effects on similar functional pathways, the specific genes/pathways on which they act are different. This is advantageous from a model perspective since it is widely believed that *T. gondii* represents a risk factor rather than a causal factor in the onset of SZ. Only a small number of statistically significant interactions, with the ability to complicate interpretation, were identified.

It has been suggested that *T. gondii* impact neurodegenerative diseases through modulating the activity of neurotransmitters, particularly dopamine. Such increases may be driven through the expression of aromatic acid hydroxylase genes by *T. gondii*, which appear similar to mammalian tyrosine hydroxylase, a rate-limiting enzyme in dopamine synthesis [22,66]. Given we were unable to detect significant expression of *T. gondii* genes in our samples, we are unable to

comment on the likelihood that exogenous manipulation of dopamine by *T. gondii* may contribute directly to the behavioral effects observed. However, we have demonstrated that chronic infection can be established in mice following a high-dose injection, leading to observable tissue cysts that were visualized in this study by imaging the accumulation of the GFP-tagged *T. gondii*. In addition, cocaine is said to elicit SZ-like symptoms in mice and acts through the dysregulation of dopamine [28]; a paradigm we leveraged for our analysis.

Another contributing mechanism that has been proposed is the disruption of tryptophan metabolism. *T. gondii* is a tryptophan auxotroph and like many micro-organisms must scavenge this amino acid from their host. The immune system has evolved a method to degrade tryptophan which is effective in restricting the growth of many such pathogens. This is achieved through the production of indoleamine 2,3-dioxygenase (IDO) in immune cells, which degrades L-tryptophan to N-formylkynurenine [18,45,67]. However, this has the effect of increasing the production of other catabolites such as 3-hydroxykynurenine and, anthranilic, xanthurenic, quinolinic and kynurenic acids. Some of these affect neurons or their functions. For example, quinolinic acid is an agonist of the N-methyl-D-aspartate (NMDA) receptor whereas both 3-OH-kynurenine and quinolinic acid can cause neuronal death by a mechanism mediated by reactive oxygen species [68]. In addition, high concentrations of kyneurenic acid (KYNA) have been detected in the cerebrospinal fluid of schizophrenic patients and may contribute to SZ associated cognitive deficiencies [45]. Furthermore, KYNA pathway markers have recently been shown to predict slowing of startle latency (a proxy for neural processing speed), with SZ patients seropositive for *T. gondii* being slower than SZ patients without evidence of latent infection [69]. Consistent with this previous study, we find that two adjacent enzymes in this pathway, kynureninase (KYNU) [EC:3.7.1.3] and 3-hydroxyanthranilate 3,4-dioxygenase (HAAO) [EC:1.13.11.6] are over-expressed in response to *T. gondii*, suggesting that enhanced Tryptophan catabolism is at play in our model.

This study demonstrates the need for further development of context dependent analyses given the limitation (loss of statistical power) that multiple-testing correction imposes on functional enrichment when using large, structured ontologies such as the Gene Ontology. We observed significant masking of enriched, neurologically relevant pathways due to the presence of a large, expected, general immune response. We have presented a simple but effective generalizable methodology whose principle should find widespread applicability in disease-specific studies.

While this study highlights some exciting associations between *T. gondii* infection and pathways involved in neurocognitive disorders, we nevertheless note several limitations in our study. First, we were unable to monitor the expression of genes by the parasite. Despite being relatively slow growing, the bradyzoite form of the parasite expresses and secretes several proteins that can impact host tissue function. Second, many genes lack functional annotations which limits our ability to resolve functions at the level of individual conditions, as well presenting challenges in the recovery of significant terms related to neurophysiology. Consequently, we did not exclude any evidence categories from the GO analysis in our assessments leading to the inclusion of lower confidence predictions. Third, we did not monitor levels of neurotransmitters which may otherwise have allowed us to correlate their expression with changes in gene expression. Fourth, the presence of complex *T. gondii* by cocaine interactions though limited, while interesting, are challenging to interpret. Fifth, we applied a liberal cutoff (FDR < 0.1) to define DE genes to ensure that our gene set captures a broad view of gene expression patterns and avoids being overwhelmed by immune response pathways. While we acknowledge that this approach increases the number of false positives, we note this approach allowed the recovery of neurophysiologically relevant terms that would otherwise be masked by immune-related terms. Finally, due to the scope of this proof-of-concept study, we chose to

study male progeny to maintain sufficient statistical power for both the behavioral and expression analysis. We note that sex differences have been reported in cocaine sensitization [70] with females being more susceptible to incentive-sensitization than males. With respect to behavioral changes due to *T. gondii* infection, gender-specific changes have been noted in several human studies e.g., [11,71] where notably, differing effects on testosterone have been reported. In mice, differences in neurotransmitter levels were reported between males and females in response to *T. gondii* infection [72]. However, these results were measured in the acute phase of infection. Some gender specific differences in gene expression have been reported though these did not appear to affect behavior [73]. Due to the possibility that there are sex-specific differences in response to latent infection, it will be important to replicate our findings across both sexes in follow-up work.

## Conclusions

We present a novel mouse model to explore molecular mechanisms underlying neurological risk factors associated with chronic *Toxoplasma gondii* infection. We have shown that chronic infection by the parasite, in combination with cocaine, produces clear alterations in behavioral paradigms that are commonly impaired in neurological disorders. Numerous behavioral changes have previously been observed following *Toxoplasma gondii* infection. Here we show a novel behavioral phenotype, that is blunted behavioral sensitization to the stimulant effects of cocaine that strengthens the link between infection and neuropsychiatric related behaviors. In addition, we identified 4439 genes that were significantly differentially expressed under these conditions. Through a systematic, curation approach we found nearly half of these genes were enriched for potentially neurologically relevant GO biological processes, highlighting potentially relevant pathways. We further demonstrated, using a more focused, novel context enrichment approach, a rapid screening technique that identified 641 neurologically relevant genes, including 114 that were not picked in the initial GSEA due to its low statistical power. Through demonstrating the capacity of this model to yield a wealth of new data on genes and pathways implicated to respond to infection by *T. gondii*, we expect further use of this model will allow the dissection of the molecular pathways by which infection with *T. gondii* contributes to disease pathogenesis enabling more efficient therapy and prevention of schizophrenia.

## Materials and methods

### Ethics statement

All procedures were approved and conducted in accordance with policies of the Hospital for Sick Children Animal Care Committee (Assigned Protocol #: 25360) and conformed to both Canadian Council on Animal Care (CCAC) and National Institutes of Health (NIH) Guidelines on Care and Use of Laboratory Animals.

### Experimental animals

Mice used in this study were the male F1 progeny of a cross between C57BL/6NtacfBr [C57B6] and 129Svev [129] mice (Taconic Farms, Germantown, NY). Mice were bred in our colony at The Hospital for Sick Children and maintained on a 12 h light/dark cycle with access to food and water *ad libitum*. A total of 39 mice were used for behavioral experiments with *T. gondii* CEP, 40 mice were used for behavioral experiments with *T. gondii* VEG and an additional cohort of mice were used for the RNA-seq experiments, a subset of which were infected with *T. gondii* CEP; one mouse treated with *T. gondii* CEP and cocaine died prior to sampling.

### *T. gondii* culture

Two type III strains were used in this study: 1) strain VEG; and 2) a CEP strain which had been engineered to contain a stably integrated GFP cassette [74]. Strains were propagated by passage in monolayers of primary human foreskin fibroblast cells (ATCC SCRC-1041), in Dulbecco's modified Eagle's medium (Wisent 319-015-CL) supplemented with 10% cosmic calf serum (HyClone SH30087.04), 20% M199 (Wisent 316-010-CL) and either penicillin/streptomycin (Wisent 450-201-EL) or, 10 U/mL penicillin and 10 μg/ml streptomycin (Gibco, Thermo Fisher Scientific Inc., Grand Island, NY), as previously described [75]. Genomic material was extracted, and PCR tested to ensure the lines were free from Mycoplasma infection. Tachyzoite-stage parasites were collected in 14 ml phosphate buffered saline (PBS) for injection into mice.

### Infection

Adult male mice, 60 days of age were injected intraperitoneally (IP) with 150 μl of PBS containing approximately 500,000 tachyzoites from either strain of *T. gondii* (GFP tagged CEP or VEG). Control mice were injected with PBS only. Mice were given an 8-week recovery period prior to behavioral testing and genomic analysis, during which the acute phase of infection was expected to subside, leading to chronic infection and associated tissue cyst formation.

### Behavioral experiments

Following the 8-week recovery period after injection, mice were assessed for fear and anxiety related behaviors (contextual fear conditioning, elevated plus maze, open field), motor ability (Rotarod) and neuropsychiatric related behaviors (including cocaine sensitization and prepulse inhibition). Behavioral procedures were conducted during the light phase of the cycle. Experiments were conducted blind to the treatment condition of the mouse, and according to local animal care protocols.

**Open field.**   A square open Field (40 cm x 40 cm) was used as a measure of anxiety and locomotor activity. The open field was placed in a room with consistent overhead lighting (250 lux). Mice were placed into the open field and their locomotor activity was recorded over a period of 5 minutes using an overhead camera connected to an automated tracking system (Limelight software, Actimetrics). Distance travelled as well as location in the open field (i.e. outer, middle versus inner zones) was measured.

**Cocaine sensitization.**   For cocaine sensitization experiments, mice were first habituated to an open arena for three days (5 minutes per day) to ensure that there was no baseline difference in locomotor activity. Then, mice were injected with either saline or 20mg/kg of cocaine daily for 5 days. Cocaine or saline was injected 5 minutes before being placed in an open field (40 cm x 40 cm). Changes in locomotor activity in response to cocaine was used as a measure of behavioural sensitization and was recorded over a period of 5 minutes using an overhead camera connected to an automated tracking system (Limelight software, Actimetrics). Dim lighting (40 lux) was used during all sessions to promote exploration of the open field. One week after the end of the sensitization the mice were tested in the same open field with locomotor activity recorded. All mice were first tested in a drug free state followed by a 20mg/kg cocaine challenge 24 hours later.

**Prepulse inhibition (PPI).**   SR-Lab startle chambers (San Diego Instruments) were used to test PPI. Mice were first placed in individual plexiglass cylinders (internal diameter 3.2 cm) which were then placed on a sensor platform inside a sound-attenuating chamber. A piezoelectric accelerometer on the base of the platform was used to detect all movements of the mouse inside the cylinder. Background white noise (65dB) was presented throughout the testing

session. After 2 minutes of acclimation to the chamber the mice were presented with a series of trials of either startle stimulus alone (40 ms duration, 120dB tone), prepulse stimulus alone (20 ms tone) or prepulse and startle stimulus pairings. The mice were presented with 5 different prepulse intensities (69 dB, 73 dB, 77 dB, 81 dB or 85 dB). For each prepulse intensity there were 12 prepulse only trials and 12 prepulse/startle trials. There were also 24 startle-only trials. Trial types were intermixed and an intertrial time of 15 s was used.

**Elevated plus maze.**    The elevated plus maze consists of a plus-shaped apparatus with 2 open and 2 enclosed arms (10 cm wide, 50 cm long), each with an open roof, elevated 1 m from the floor. Mice were placed in the center of the maze and allowed to explore for 5 mins. Their movements were tracked by an overhead camera and analyzed using Any-maze software. The proportion of time spent in the closed arms was calculated (time in closed/total time) and used as a measure of anxiety. In EPM, this translates into a restriction of movement to the enclosed arms.

**Contextual fear conditioning.**    Contextual fear conditioning was performed in Med-associates conditioning chambers (31 cm x 24 cm x 21 cm) containing shock-grid floors. During training mice were placed in the chamber and were allowed to acclimatize for 2 minutes. Then, they received 3 foot-shocks (0.5 mA, 2 s duration spaced 1 minute apart) and were removed from the chamber 1 minute after the final shock. The following day, mice were placed back in the same conditioning chamber for 5 minutes. No shocks were delivered, and the time spent freezing (defined as the absence of all movement other than respiration) was recorded using Freeze Frame analysis software (Actimetrics).

**Rotarod.**    Motor learning was assessed using an accelerating protocol on an automated Rota-rod system (Med-Associates). Mice were trained for 5 days with 4 trials per day. The maximum trial length was 5 minutes during which time the speed of the rotation accelerated linearly from 4 to 40 revolutions per minute. The length of time the mice stayed on the rod before falling was recorded for each trial. Mice were given an approximate intertrial interval of 10 minutes.

**Tissue collection.**    The hippocampus and striatum were dissected from mouse brains, frozen in dry ice cooled isopentane and stored at -80C. Frozen tissues were homogenized through bead-beating in RLT lysis buffer (Qiagen Rneasy kit, cat. 74104) using a single pre-chilled 5mm steel bead (Qiagen cat. 69989) and the Qiagen TissueLyser for 2x 2 min at 30 Hz, followed by passage through QIAshredder spin columns (Qiagen cat. 79654).

## Gene expression

**RNA extraction and sequencing.**    Total RNA extraction was carried out using the Qiagen Rneasy kit (cat. 74104) with on-column *Dnase*I treatment, as per the manufacturer's instructions. Quality of RNA samples was checked on an Agilent Bioanalyzer 2100 RNA Nano chip following Agilent Technologies' recommendation. Concentration was measured by Qubit RNA HS Assay on a Qubit fluorometer (ThermoFisher). Yields ranged from 8.5 to 21.9 μg. Only samples with RNA integrity number > 6.9 and little to no degradation apparent on electrophoretograms were accepted. Library preparation and sequencing were performed at The Centre for Applied Genomics (TCAG; Toronto, ON). Library preparation was performed following the NEBNext Ultra II Directional RNA Library Preparation protocol (New England Biolabs, Ipswich, MA). Briefly, 800 ng of total RNA was enriched for poly-A mRNA, fragmented into the 200-300-bases range for 4 minutes at 94˚C and converted to double stranded cDNA, end-repaired and adenylated to create a 3' overhang for ligation of Illumina adapters. Following PCR amplification and extension, each sample contained a different barcoded adapter to allow for multiplex sequencing. One ul of each of the RNA-seq libraries was loaded

on a Bioanalyzer 2100 DNA High Sensitivity chip (Agilent Technologies) to check for size and absence of primer dimers. RNA libraries were quantified by qPCR using the Kapa Library Quantification Illumina/ABI Prism Kit protocol (KAPA Biosystems), pooled in equimolar quantities and single-end sequenced on a HiSeq 2500 platform using v4 chemistry following Illumina's recommended protocol to generate single-end reads of 100-bases in length (40 million reads per sample; 250–270 million reads per lane).

**Pre-processing and annotation.**  We first processed sequence reads by removing adaptor and vector contaminated reads and trimmed low quality reads using Trimmomatic v.0.36 [76]. Quality reads were mapped to the reference genomes *M. musculus* (Source: Ensembl; Taxonomic ID: 10090, Release 87) and *T. gondii* (Source: Ensembl; Taxonomic ID: 432359, Release 34) using STAR [38]. Expression levels of mouse genes were normalized across samples with reads per kilobase per million reads (RPKM).

**Statistical analysis.**  To assess the effects of *T. gondii* infection, cocaine treatment, and cocaine-*T. gondii* interactions, DE testing was performed using likelihood ratio tests on different negative binomial general linear models of gene expression. For *T. gondii* main effects, a likelihood ratio test was done for each gene comparing the performance of the model "gene expression ~ *T. gondii* + cocaine" against the reduced model "gene expression ~ cocaine". To assess cocaine main affects, a full model of "gene expression ~ *T. gondii* + cocaine" was compared to the reduced model "gene expression ~ *T. gondii*". Finally, to assess the significance of *T. gondii* by cocaine interactions, a full model of "gene expression ~ *T. gondii* + cocaine + *T. gondii* ´ cocaine" was compared to the reduced model "gene expression ~ *T. gondii* + cocaine". False discovery rate (FDR) correction was performed on p-values across all genes separately within each different likelihood ratio test (i.e., *T. gondii* main effect, cocaine main effect, and *T. gondii* by cocaine interaction), with an FDR cutoff of 10% taken for statistical significance. Analyses were performed separately on the striatal and hippocampal samples. All gene differential expression analyses were performed using the DESeq2 R Package version 1.22.1 and R version 3.5.3 [77]. Count normalization size factors were calculated across all samples using the median geometric mean-scaled count method (default method). These 'global' size factors were then used in all analyses. DE results were grouped into 4 main categories: 1) significant *T. gondii* by cocaine interactions, 2) significant main effect of *T. gondii*, significant main effect of cocaine, and no significant *T. gondii* by cocaine interaction, 3) significant main effect of *T. gondii* only (no significant main effect of cocaine and no significant *T. gondii* by cocaine interaction) and 4) significant main effect of cocaine only (no significant main effect of *T. gondii* and no significant *T. gondii* by cocaine interaction). Groups 2–4 were further broken down based on whether *T. gondii* and cocaine main effects consisted of upregulation or downregulation in response to *T. gondii* or cocaine (**S1 Data).** Genes with significant interactions were clustered into unique interaction types using log2 fold-change signed p-values from pairwise Wald differential abundance tests between groups. Heatmaps of significant genes were generated using mean-centered and standard deviation-scaled log2 values of the DESeq2 size factor normalized counts (See **S4**–**S11 Figs** and **S3 and S4 Data**).

Fisher's Exact Tests were performed in R to determine whether significant differences existed between the expected and observed number of *T. gondii* and cocaine responsive genes in hippocampus and striatum (**S4 Data**) and to determine whether there were significant differences in the number of differentially expressed genes within gene subsets or conditions for each tissue. The expected values for these were calculated based on the null hypothesis that condition had no effect on the totals. Thus, the expected values in each category reflect the relative frequency of hippocampus-specific and striatum-specific genes for all differentially expressed genes.

**Functional analysis.** Gene set enrichment analysis was performed on DE genes with the Cytoscape plugin, BiNGO [43,78] using the Benjamini-Hochberg false discovery rate procedure (FDR<0.05) for multiple testing correction (**S5 Data**). Additional gene set enrichments were performed in BiNGO by constructing custom ontologies using Reactome and KEGG [79]. All ontology files and annotations used for this analysis are provided as **S6 Data**. Due to the large occurrence of generic, high-level immune system processes, significant GO terms were curated to assess their potential neurological relevance according to the method below (see Curation). Data files were imported for visualization in RStudio and plotted using the R libraries: UpSetR, rJava, grid, tidyverse and venneuler. Intersects for genes in term groups were plotted in R using UpSetR with nintersects = 45.

Context enrichment was performed as follows. We first defined a set of root keywords ("neuro", "brain", "behavior", "locomot", "memory", and "synap") which we considered, based on our curation of the literature, and in the context of our experiments to be both relevant and non-immune system specific. We then performed an AmiGO [54] search for GO biological process terms containing these keywords among their full-text descriptions (see **S2 Fig** and **S8 Data**). The resulting 1632 GO terms defined the functional "context" and were used to filter the DE gene set by removing from consideration those that did not contain at least one annotation matching the defined context. GO enrichment analysis was then performed on this subset per above using both the full GO ontology and NIGO, a neuro-immune focused GO subset developed using a clipping approach based on domain-specific relevance [56] obtained from BioPortal [80]. We obtained the list of genes with differentially methylated positions associated with psychosis, schizophrenia, and treatment-resistant schizophrenia from supplementary material in Hannon et al. [55]. The expected overlap between this set and our context-enriched set was assessed using Fisher's Exact Test assuming an estimated background frequency of 658/21,306 for SZ-associated methylated mouse genes.

**Data curation.** Neurological relevance of statistically enriched GO terms occurring in DE gene annotations was confirmed by curation as follows. First, terms containing words indicative of neuronal function e.g., brain development, behavior, neuroinflammation or brain-specific cells or processes, were included. Second, using the form *pathway* + "in brain" e.g., "NF-kappaB in brain", we conducted Google searches (i.e., page rank) and manually reviewed scholarly sources included in first page results for supporting evidence of neurologically associated function. Scholarly sources were defined as primary scientific literature accessible on PubMed or, reputable aggregate sources (e.g., GeneCards [81], RefSeq [82] or, Uniprot [83]) citing primary sources. Related terms were grouped based on GO hierarchy with each term being represented only once. For each experimental condition, a '1' was scored where at least one neurologically relevant term in a given group was enriched (see **S7 Data**).

## Supporting information

**S1 Fig. Behavioral testing of *T. gondii* (VEG) infected mice.** A—Infected mice are not impaired in a contextual fear conditioning (t-test, $p = 0.28$) B-C- Infected mice did not show altered anxiety behaviors in the elevated plus maze (Two-way ANOVA No significant main effect of group ($p = 0.83$) or region by group interaction ($p = 0.92$) (B) or open field test (Two way ANOVA No significant main effect of group ($p = 0.99$) or group by zone interaction ($p = 0.34$)) (C). D- infected mice exhibited a significant reduction in distance travelled in the open field (t-test $t(18) = 2.24$, $p = 0.038$). E—There was also a significant decrease in the time spent on the rotarod compared to control mice (Repeated measures ANOVA, main effect of group F$(4,90) = 5.1$, $p = 0.026$. group by session interaction $p = 0.89$. F–A reduction in prepulse inhibition was observed in infected mice compared to controls (Repeated measures ANOVA main

effect of group F(4,90) = 31.69, $p \leq 0.0001$ group by intensity interaction $p = 0.45$). G-H–As was observed with the CEP strain, infected mice had a blunted locomotor response to repeated cocaine administration significant group by drug treatment effect F(1,80) = 8.09, $p = 0.0057$) (G) however, there was no significant group or group type interactions (*p's* $\geq 0.14$*)* during a subsequent cocaine challenge one week following sensitization (H).
(PDF)

**S2 Fig. Numbers of GO terms returned in neurological keyword searches with overlaps.**
(PDF)

**S3 Fig. Expression patterns of interacting DE genes by functional effect group (A to H).** Significant differences are indicated by an asterisk (Chi square; p < 0.05).
(PDF)

**S4 Fig. Heatmap of gene expression defining DE gene responses by type and subtype.** Hippocampus– 40 genes with significant (FDR<0.1) *T. gondii* by cocaine interactions.
(PDF)

**S5 Fig. Heatmap of gene expression defining DE gene responses by type and subtype.** Hippocampus, 364 genes with significant (FDR<0.1) *T. gondii* and cocaine effects.
(PDF)

**S6 Fig. Heatmap of gene expression defining DE gene responses by type and subtype.** Hippocampus, 1143 genes with significant (FDR<0.1) T. gondii (only) effects.
(PDF)

**S7 Fig. Heatmap of gene expression defining DE gene responses by type and subtype.** Hippocampus, 1365 genes with significant (FDR<0.1) cocaine (only) effects.
(PDF)

**S8 Fig. Heatmap of gene expression defining DE gene responses by type and subtype.** Striatum, 165 genes with significant (FDR<0.1) *T. gondii* by cocaine interactions.
(PDF)

**S9 Fig. Heatmap of gene expression defining DE gene responses by type and subtype.** Striatum, 173 genes with significant (FDR<0.1) *T. gondii* and cocaine effects.
(PDF)

**S10 Fig. Heatmap of gene expression defining DE gene responses by type and subtype.** Striatum, 1819 genes with significant (FDR<0.1) *T. gondii* (only) effects.
(PDF)

**S11 Fig. Heatmap of gene expression defining DE gene responses by type and subtype.** Striatum, 920 genes with significant (FDR<0.1) cocaine (only) effects.
(PDF)

**S1 Data. Supplementary spreadsheet (macro-enabled) containing colorized heatmap of significant, differentially expressed genes in mouse hippocampus and striatum with accompanying metadata.**
(XLSM)

**S2 Data. Supplementary spreadsheet detailing the statistical breakdown of DESeq2 significant gene sets in hippocampus.**
(XLSX)

**S3 Data. Supplementary spreadsheet detailing the statistical breakdown of DESeq2 significant gene sets in striatum.**
(XLSX)

**S4 Data. Supplementary spreadsheet with statistical analysis of genes in categories**
(XLSX)

**S5 Data. Zip archive containing functional enrichment details (i.e., BiNGO output).**
(ZIP)

**S6 Data. Zip archive of ontologies and annotations used in this study.**
(ZIP)

**S7 Data. Supplementary spreadsheet with curation of neurologically relevant functional terms.**
(XLSX)

**S8 Data. Supplementary spreadsheet with context enrichment analysis.**
(XLSX)

**S9 Data. Cytoscape session file with functional annotations associated with context-enriched genes.** This file is in a format required by the Cytoscape software, an open source platform for visualizing complex networks (https://cytoscape.org).
(CYS)

**S10 Data. Supplementary spreadsheet with analysis of significant CxT interactions.**
(XLSX)

## Acknowledgments

We would like to thank Dr. Steven Maere, Ghent University for providing helpful advice on the creation of custom annotation files for use in BiNGO.

## Author Contributions

**Conceptualization:** Jonathan R. Epp, Paul W. Frankland, John Parkinson.

**Data curation:** Graham L. Cromar, Ana Popovic, Xuejian Xiong.

**Formal analysis:** Graham L. Cromar, Jonathan R. Epp, James St. Pierre, Xuejian Xiong.

**Funding acquisition:** Jonathan R. Epp, Paul W. Frankland, John Parkinson.

**Investigation:** Graham L. Cromar, Jonathan R. Epp, Ana Popovic, Yusing Gu, Violet Ha, Brandon J. Walters, Xuejian Xiong.

**Methodology:** Jonathan R. Epp, Brandon J. Walters, John G. Howland, Sheena A. Josselyn, Paul W. Frankland, John Parkinson.

**Supervision:** Paul W. Frankland, John Parkinson.

**Visualization:** Graham L. Cromar, Jonathan R. Epp, James St. Pierre, John Parkinson.

**Writing – original draft:** Graham L. Cromar, John Parkinson.

**Writing – review & editing:** Graham L. Cromar, Jonathan R. Epp, Sheena A. Josselyn, Paul W. Frankland, John Parkinson.

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
