## [Decision Letter · Decision Letter 0]

9 May 2022

Dear Dr. Parkinson,

Thank you very much for submitting your manuscript "Toxoplasma infection in male mice alters dopamine-sensitive behaviors and host gene expression patterns associated with neuropsychiatric disease" for consideration at PLOS Neglected Tropical Diseases. As with all papers reviewed by the journal, your manuscript was reviewed by members of the editorial board and by several independent reviewers. The reviewers appreciated the attention to an important topic. Based on the reviews, we are likely to accept this manuscript for publication, providing that you modify the manuscript according to the review recommendations. 

The reviewers felt that the manuscript was well-written and that results presented are of “paramount importance”, and at this time only minor revisions are necessary. 

Some important revisions are suggested, including clarifying some results in the figures and some edits to the discussion. Importantly, reviewer 3 considered it a “major reservation” that more comparisons should be made to prior data fro the literature, and made some suggestions for interpretation of the data from Figure 2. Reviewer 1 additionally made several helpful suggestions for improving the structure and presentation of the figures. Please considered these suggested revisions.

Sincerely,

Bruce A. Rosa

Associate Editor

Anthony Papenfuss

Deputy Editor

The reviewers felt that the manuscript was well-written and that results presented are of “paramount importance”, and at this time only minor revisions are necessary. 

Some important revisions are suggested, including clarifying some results in the figures and some edits to the discussion. Importantly, reviewer 3 considered it a “major reservation” that more comparisons should be made to prior data fro the literature, and made some suggestions for interpretation of the data from Figure 2. Reviewer 1 additionally made several helpful suggestions for improving the structure and presentation of the figures. Please considered these suggested revisions.

Reviewer's Responses to Questions

**Key Review Criteria Required for Acceptance?**

**Methods**

-Are the objectives of the study clearly articulated with a clear testable hypothesis stated?

-Is the study design appropriate to address the stated objectives?

-Is the population clearly described and appropriate for the hypothesis being tested?

-Is the sample size sufficient to ensure adequate power to address the hypothesis being tested?

-Were correct statistical analysis used to support conclusions?

-Are there concerns about ethical or regulatory requirements being met?

Reviewer #1: It is not clear in the methods or figure legend what the behavioral sensitization paradigm test is performed for figure 2 panels G and H. It is especially confusing because it is labeled the same as the open field test in panel E but no differences were seen between T. gondii infected and uninfected mice treated with vehicle only. It is unclear why there are no differences in this test in response to T. gondii infection.

Reviewer #2: (No Response)

Reviewer #3: The methods are clear and well described for reproduction and assessment relative to other published studies. The animals used are clearly described and ethical considerations followed. The sample size is sufficient for this analysis.

**Results**

-Does the analysis presented match the analysis plan?

-Are the results clearly and completely presented?

-Are the figures (Tables, Images) of sufficient quality for clarity?

Reviewer #1: 1. The dots on figure 3 are very confusing. Are the dots supposed to align with the conditions on the left? The number of affected genes do not match up. For example, is the first dot for Toxo Up? If so why are the number of affected genes in the hippocampus the same (837 vs 1150). Maybe a Venn Diagram would be clearer? Then the reader could easily see how many genes are shared between the various conditions. 

2. For figure 5, the term descriptions are not that long, so it would be better to have them up in the figure instead of just having the letter code. It is difficult for the reader to go back and forth with the key if want to look at several term descriptions.

3. The term “combined” referring to T. gondii and cocaine together is very confusing in figure 6 as it is not described in later in the text. From the figure legend it reads that combined is hippocampus and striatum. If you want to use the term combined define it clearly in the figure legend and in the results test right away, but it would likely be best to just write Cocaine/T. gondii instead of combined.

Reviewer #2: (No Response)

Reviewer #3: The analysis in the Results fits the experimental plan and adequate figures are presented. Some specific comments on presentation are below. 

There are some concerns about the data presented is the General Comments below.

**Conclusions**

-Are the conclusions supported by the data presented?

-Are the limitations of analysis clearly described?

-Do the authors discuss how these data can be helpful to advance our understanding of the topic under study?

-Is public health relevance addressed?

Reviewer #1: discussion and conclusions are fine

Reviewer #2: (No Response)

Reviewer #3: The Conclusions are extensive but accurate with in depth interpretation presented in light of published work. The relevance is clear and its contribution to our understanding.

**Editorial and Data Presentation Modifications?**

Reviewer #1: (No Response)

Reviewer #2: (No Response)

Reviewer #3: Lines 91-93 Rewrite sentence because the neurological effects during AIDS may be due to cyst reactivation and active brain cell lysis. 

line 98-103 Include the NE changes found two studies that impacts cytokine levels in the CNS. 

Line 129-138 Delete from Intro as these are Results.

Delete Fig 5A and 6B. Pleasing to the eye but better represented by tables or graphs with error bars and values. 

Line 417-434 This last analysis can be deleted.

**Summary and General Comments**

Reviewer #1: This manuscript details the authors development of a novel mouse model to examine the effects of cocaine treatment on mice with a chronic infection of the parasite Toxoplasma gondii. They perform the series of behavioral tests and find that T. gondii infection blunts the stimulatory effects of cocaine treatment. They then performed RNAseq of the hippocampus and striatum of mice with and without T. gondii chronic infection, and with and without cocaine treatment. Extensive expression and GO analyses are performed on these datasets. While no direct follow-up studies were performed yet, interesting functional effect groups were identified for future studies. Overall the manuscript is extremely well written but the figure and legends are not always clear, so suggestions were made for improvement.

Reviewer #2: This manuscript (MS) reports interesting findings that Toxoplasma infection in male mice alters dopamine-sensitive behaviors and host gene expression patterns associated with neuropsychiatric disease. This novel mouse model may present a new perspective to elucidate the molecular pathways by which T. gondii infection contributes to neuropsychiatric disorders. 

This study was properly designed and the experiments were meticulously performed. The MS is well-written, and the conclusions are supported by the presented data. This MS should be accepted in its present form.

Reviewer #3: Overall a well thought out and conducted study providing clarity on a complex and interesting topic. A novel approach is applied that could benefit both SZ research and understanding chronic Toxoplasma infection and its link with SZ. 

A major reservation that needs addressing in revised versions is comparing with prior data. 

In Fig 2, the increased time in open zones and increased movement with infection, as found in other studies, is not reproduced. The decreased distance travelled in infected mice may be due to sickness phenotypes and hence wellness and fitness of infected animals should be reported. There is some concern that sickness phenotypes could account for the lack of response for cocaine in Fig 2G. Particularly with the extremely high dose of infection used. Further, could the differences from published behaviour studies be accounted for by Tg strain differences since these are Type III?

PLOS authors have the option to publish the peer review history of their article (what does this mean?). If published, this will include your full peer review and any attached files.

Reviewer #1: Yes: Laura J. Knoll

Reviewer #2: No

Reviewer #3: No

Figure Files:

Data Requirements:

Reproducibility:

References

---

## [Editor Report · Decision Letter 1]

15 Jun 2022

Dear Dr. Parkinson,

Thank you very much for submitting your manuscript "Toxoplasma infection in male mice alters dopamine-sensitive behaviors and host gene expression patterns associated with neuropsychiatric disease" for consideration at PLOS Neglected Tropical Diseases. 

At this time I feel that you have adequately addressed the reviewer comments with a number of revisions throughout the manuscript, but I have some minor formatting change suggestions before finalizing the manuscript. 

Regarding reviewer 1's suggestion to replace the upset plot with a Venn diagram, keeping the upset plot is acceptable. Venn diagrams are more straightforward to understand and interpret, but upset plots do provide additional information and data summaries, as the authors have indicated. The additional description added the caption in the revised version should help readers who are not familiar with this type of plot. 

Regarding reviewer 3's suggestion about removing the two figure panels, I understand the reviewer's concerns, but I agree with the authors that they warrant being included. For Figure 5A, there is no logical replacement for the word cloud in terms of a table or a graph, since it's just intended as a visual summary of the overall pathway data, and not as a quantitative or statistical approach. It is effective at communicating the intended data. For Fig 6B, as the authors indicate, the branch data can help to interpret findings for specific pathways in the parent-child context, and also help to group similar terms together which can make the data easier to digest. The color and size of the nodes indicate P values and gene number (respectively) so there is not data hidden using this approach.

I suggest a few other minor changes: 

1. For Figure 4B, the x-axis labels are not very neatly aligned. The last four labels line up close to the center of the blue and purple bars, but the first four align after the bars. All of these labels could be moved up closer to the axis as well. 

2. For Figure 5B, now that the full names of the terms have been added, could the letter indicators (j., t., f., a., etc) before the term names be removed? I don't see any use for them and it would clean the figure up. I also suggest that the P value ranges indicated above and below the colored nodes be moved to the end of the figure caption instead of being on the figure itself. Those two changes will improve readability of the figure. 

3. Since the supplementary methods are just a single page describing statistical tests used, this should be moved to the main text methods instead. 

Sincerely,

Bruce A. Rosa

Associate Editor

Anthony Papenfuss

Deputy Editor

At this time I feel that you have adequately addressed the reviewer comments with a number of revisions throughout the manuscript. 

- Regarding reviewer 1's suggestion to replace the upset plot with a Venn diagram, keeping the upset plot is acceptable. Venn diagrams are more straightforward to understand and interpret, but upset plots do provide additional information and data summaries, as the authors have indicated. The additional description added the caption in the revised version should help readers who are not familiar with this type of plot. 

- Regarding reviewer 3's suggestion about removing the two figure panels, I understand the reviewer's concerns, but I agree with the authors that they warrant being included. For Figure 5A, there is no logical replacement for the word cloud in terms of a table or a graph, since it's just intended as a visual summary of the overall pathway data, and not as a quantitative or statistical approach. It is effective at communicating the intended data. For Fig 6B, as the authors indicate, the branch data can help to interpret findings for specific pathways in the parent-child context, and also help to group similar terms together which can make the data easier to digest. The color and size of the nodes indicate P values and gene number (respectively) so there is not data hidden using this approach.

I suggest a few other minor changes: 

1. For Figure 4B, the x-axis labels are not very neatly aligned. The last four labels line up close to the center of the blue and purple bars, but the first four align after the bars. All of these labels could be moved up closer to the axis as well. 

2. For Figure 5B, now that the full names of the terms have been added, could the letter indicators (j., t., f., a., etc) before the term names be removed? I don't see any use for them and it would clean the figure up. I also suggest that the P value ranges indicated above and below the colored nodes be moved to the end of the figure caption instead of being on the figure itself. Those two changes will improve readability of the figure. 

3. Since the supplementary methods are just a single page describing statistical tests used, this should be moved to the main text methods instead. 

Figure Files:

Data Requirements:

Reproducibility:

References

---

## [Editor Report · Decision Letter 2]

21 Jun 2022

Dear Dr. Parkinson,

We are pleased to inform you that your manuscript 'Toxoplasma infection in male mice alters dopamine-sensitive behaviors and host gene expression patterns associated with neuropsychiatric disease' has been provisionally accepted for publication in PLOS Neglected Tropical Diseases.

Best regards,

Bruce A. Rosa

Associate Editor

Anthony Papenfuss

Deputy Editor

The authors have addressed all reviewer concerns, and the manuscript will now be considered accepted.

---

## [Editor Report · Acceptance letter]

13 Jul 2022

Dear Dr. Parkinson,

We are delighted to inform you that your manuscript, "*Toxoplasma* infection in male mice alters dopamine-sensitive behaviors and host gene expression patterns associated with neuropsychiatric disease," has been formally accepted for publication in PLOS Neglected Tropical Diseases.

Best regards,

Shaden Kamhawi

co-Editor-in-Chief

Paul Brindley

co-Editor-in-Chief
